# Xylanase modulates the microbiota of ileal mucosa and digesta of pigs fed corn-based arabinoxylans likely through both a stimbiotic and prebiotic mechanism

**Amy L. Petry**[1], **John F. Patience**[1,2], **Lucas R. Koester**[3], **Nichole F. Huntley**[1], **Michael R. Bedford**[4], **Stephan Schmitz-Esser**[1] *

**1** Department of Animal Science, Iowa State University, Ames, Iowa, United States of America, **2** Iowa Pork Industry Center, Iowa State University, Ames, Iowa, United States of America, **3** Department of Veterinary Microbiology and Preventive Medicine, Iowa State University, Ames, Iowa, United States of America, **4** AB Vista Feed Ingredients, Marlborough, Wiltshire, United Kingdom

\* sse@iastate.edu

**Data Availability Statement:** All relevant data and 16S rRNA gene sequences are available at the National Center for Biotechnology Information Sequence Read Archive under the BioProject ID:

## Abstract

The experimental objective was to characterize the impact of insoluble corn-based fiber, xylanase, and an arabinoxylan-oligosaccharide on ileal digesta and mucosa microbiome of pigs. Three replicates of 20 gilts were blocked by initial body weight, individually-housed, and assigned to 1 of 4 dietary treatments: a low-fiber control (LF), a 30% corn bran high-fiber control (HF), HF+100 mg/kg xylanase (HF+XY), and HF+50 mg/kg arabinoxylan oligosaccharide (HF+AX). Gilts were fed their respective treatments for 46 days. On day 46, pigs were euthanized and ileal digesta and mucosa were collected. The V4 region of the 16S rRNA was amplified and sequenced, generating a total of 2,413,572 and 1,739,013 high-quality sequences from the digesta and mucosa, respectively. Sequences were classified into 1,538 mucosa and 2,495 digesta operational taxonomic units (OTU). Hidden-state predictions of 25 enzymes were made using Phylogenetic Investigation of Communities by Reconstruction of Unobserved States 2 (PICRUST2). Compared to LF, HF increased *Erysipelotrichaceae_UCG-002*, and *Turicibacter* in the digesta, *Lachnospiraceae_unclassified* in the mucosa, and decreased *Actinobacillus* in both ($Q<0.05$). Relative to HF, HF+XY increased 19 and 14 of the 100 most abundant OTUs characterized from digesta and mucosa, respectively ($Q<0.05$). Notably, HF+XY increased the OTU_23_*Faecalibacterium* by nearly 6 log$_2$-fold change, compared to HF. Relative to HF, HF+XY increased genera *Bifidobacterium*, and *Lactobacillus*, and decreased *Streptococcus* and *Turicibacter* in digesta ($Q<0.05$), and increased *Bifidobacterium* and decreased *Escherichia-Shigella* in the mucosa ($Q<0.05$). Compared to HF, HF+AX increased 5 and 6 of the 100 most abundant OTUs characterized from digesta and mucosa, respectively, ($Q<0.05$), but HF+AX did not modulate similar taxa as HF+XY. The PICRUST2 predictions revealed HF+XY increased gene-predictions for enzymes associated with arabinoxylan degradation and xylose metabolism in the digesta, and increased enzymes related to short-chain fatty acid production in the mucosa. Collectively, these data suggest xylanase elicits a stimbiotic and prebiotic mechanism.

PRJNA664606 or through this link: https://www.
ncbi.nlm.nih.gov/bioproject/PRJNA664606.

**Funding:** Amy Petry (AP) is a recipient of a United
States Department of Agriculture National Institute
of Food and Agriculture (USDA-AFRI; https://nifa.
usda.gov) National Needs Fellowship for Ph.D.
Studies (Grant # 2016-38420-25496). USDA-AFRI
provided support in the form of salaries for AP
only, but the funder did not have any additional role
in the study design, data collection and analysis,
decision to publish, or preparation of the
manuscript. This study was supported by AB Vista,
and they provided the xylanase and AXOS used in
this trial. MR Bedford is an employee of AB Vista.
As one of the authors, MB had influence over the
study design. However, MB had a smaller role in
the decision to publish or preparation of the
manuscript, and there was no point at which
conclusions or interpretations put forward, that
were suggested by MB, without full agreement of
the other authors. AB Vista was involved in the
conceptualization of the project and study design.
The specific roles of all authors are articulated in
the 'author contributions' section.

**Competing interests:** The authors have read the
journal's policy and have the following competing
interests: MB is a paid employee of AB Vista. This
study evaluates a product marketed by AB Vista.
This does not alter our adherence to PLOS ONE
policies on sharing data and materials.

## Introduction

Historically, there has been a discordant view about dietary fiber (DF) among monogastric
nutritionists. In swine production, DF is often eschewed as much as possible, particularly
insoluble DF sources, due to its antinutritive effects that result in reduced pig performance
and carcass yield [1]. In contrast, companion animal and human nutritionists have concen-
trated on the health benefits of increasing DF, such as modulation of microbiota, glycemic
and appetite control, laxation, disease prevention and improved intestinal health [2]. These
different perspectives can largely be attributed to diet formulation objectives, and the
sources of DF used in these different species. The inclusion of DF is more intentional in
companion animal and human nutrition, as their aim for diet composition largely focuses
on longevity and weight maintenance, and thus, the aforementioned benefits of providing
DF favor its use. Pork production, on the other hand, tends to focus on optimizing growth
and feed efficiency at low cost, production outcomes that may be compromised by increased
DF levels. Dietary fiber is most commonly increased in swine diets through the use of indus-
trial co-products when they are used to reduce feed cost [3]. There has been renewed interest
in the benefits of microbial modulation via DF as numerous pork producers have moved
towards antibiotic-free production [4]. Still, many of the economically practicable DF
sources for swine diets are largely insoluble and poorly fermented, and thus, unable to mod-
ulate gut microbiota with the same efficacy as more fermentable DF sources [5,6]. Exoge-
nous carbohydrases may bridge the gap between the economics of using insoluble DF
sources (e.g. industrial co-products) and the health benefits of more rapidly fermentable DF
[7].

Swine diets frequently contain substantial levels of arabinoxylans, a non-starch polysaccha-
ride innate to cereal grains; often, they stem from corn and corn co-products [8]. It is well
established that pigs do not produce the exogenous enzymes needed to hydrolyze arabinoxy-
lans. Xylanase is a carbohydrase that hydrolyzes the β-(1–4) glycosidic bonds of arabinoxylans,
and thus, has the potential to partially ameliorate the antinutritive effects of DF [9]. Moreover,
an unexpected finding from utilizing xylanase in commercial pork production is its ability to
reduce finishing pig mortality [10]. Likewise, supplementing xylanase in corn-based diets has
been shown to modulate immune function [11], alter intestinal morphology [12], increase
markers indicative of improved gut barrier integrity [13], and mitigate markers of oxidative
stress [14].

It has been hypothesized that these modulations in gastrointestinal physiology are a provi-
sion of microbial modulation by the release products of xylanase, arabinoxylan oligosaccha-
rides [AXOS; 15,16]. These AXOS can be rapidly fermented by resident microbiota, and
potentially could act in a prebiotic and/or stimbiotic like manner [16,17]. Prebiotic is defined
as '*a substrate that is selectively utilized by host microorganisms conferring a health benefit*' [18],
and a stimbiotic is defined as a '*an additive that stimulates a fiber-degrading microbiome result-
ing in an increase in fiber fermentability even though the additive itself contributes little to SCFA
production*' [16]. There is a paucity of research investigating the composition of distal ileal
microbiota in pigs fed xylanase supplemented in a corn-based diet. Similarly, there is a dearth
of research investigating the impact of supplementing AXOS on the ileal microbiome of pigs.
Therefore, the experimental objective was to characterize the impact of insoluble fiber, xyla-
nase, and directly supplemented AXOS on the ileal digesta and mucosa microbiome of pigs
fed corn-based fiber. We hypothesized that the addition of xylanase or AXOS would modulate
intestinal microbial communities that support fiber degradation (stimbiotic mechanism) or
would be selectively utilized by communities that are known to confer a health benefit (prebi-
otic mechanism).

## Materials and methods

All experimental protocols were approved by the Iowa State University Institutional Animal Care and Use Committee (#9-17-8613-S) and adhered to guidelines for the ethical and humane use of animals for research according to the Guide for the Care and Use of Agricultural Animals in Research and Teaching [19].

### Experimental design

The experimental design, animal methods, and diets for this study were previously reported by Petry et al., [14] in a related study. In brief, a total of 60 growing gilts (progeny of Camborough sows × 337 sires; PIC Inc., Hendersonville, TN) were used in 3 replicates (20 gilts per replicate) of a 46-day trial. Gilts were individually housed in pens for 36 days, and subsequently moved to metabolism crates for the previously published 10-d metabolism study. Pigs were blocked by initial body weight (25.4 ± 0.9 kg) and randomly assigned within a block to 1 of 4 dietary treatments (n = 15; 5 pigs per treatment within a replicate): a low-fiber control (LF) with 7.5% neutral detergent fiber (NDF), a 30% corn bran without solubles higher-fiber control (HF; NDF = 21.9%), HF + 100 mg xylanase/kg (HF+XY; Econase XT 25P; AB Vista, Marlborough, UK) providing 16,000 birch xylan units per kg, and HF + 50 mg arabinoxylan-oligosaccharide/kg (HF+AX; 3–7 degrees of polymerization). The xylanase included in HF+XY was isolated from *Trichoderma reesei* using submerged fermentation. The AXOS supplemented in HF+AX was produced by hydrothermal treatment of corn cobs and subsequent xylanase treatment to produce oligosaccharides which range from 3–7 degrees of polymerization. Pigs were fed *ad libitum* for the 36-day adaptation period, and then limit fed 80% of the average daily feed intake among all treatments of the first replicate for the 10-day metabolism study. Pigs were given *ad libitum* access to water through the duration of the total study.

### Sample collections, DNA extraction and sequencing

On day 46, pigs were fed half of their total daily feed allotment, and after consumption, were euthanized by captive bolt stunning and exsanguination. Pigs were necropsied and an approximately 24-cm section of ileum, 20 cm proximal to the ileocecal junction, was isolated. Ileal digesta was collected from this section, snap-frozen in liquid nitrogen, and stored at -80˚C for later analysis. From this isolated section, a 12-cm portion was rinsed with sterile phosphate buffered saline, and mucosa scrapings were carefully collected, snap-frozen in liquid nitrogen, and stored at -80˚C for later analysis. Total genomic DNA was extracted from ileal mucosa and digesta using a DNeasy PowerLyzer PowerSoil Kit (Qiagen, Germantown, MD) according to the manufacturer's instructions. Extracted genomic DNA concentration and purity were evaluated using a ND-1000 spectrophotometer (NanoDrop Technologies, Rockland, DE), and subsequently stored at -80˚C for later sequencing. All samples had 260:280 nm ratios above 1.82. Extracted DNA was adjusted to a total well volume of 125 ng of DNA, and sequencing was conducted in the DNA facility at Iowa State University (Ames, IA).

A PCR-amplified 16S rRNA sequencing was conducted using a previously established protocol designed to amplify bacteria and archaea [20]. In brief, one replicate per sample of extracted genomic DNA was amplified using Platinum™ Taq DNA Polymerase (Thermo Fisher Scientific, Waltham, MA). Collective 16S rRNA bacterial primers [515F (5′-GTGYCAGCMG CCGCGGTAA-3′; 21] and 806R [5′-GGACTACNVGGGTWTCTAAT-3′; 22] for the hypervariable region V4 were utilized as previously explained by Kozich et al., [23]. All samples were subjected to PCR with an initial denaturation step at 94˚C for 3 min, followed by 35 PCR cycles (45s of denaturing at 94˚C, 20s of annealing at 50˚C, and 90s of extension at 72˚C), and concluded with a 10 min extension at 72˚C. Subsequentially, PCR products were purified with a

QIA quick 96 PCR Purification Kit (Qiagen Sciences Inc, Germantown, MD) according to the manufacturer's recommendations. The bar-coded amplicons were included at equal molar ratios and used for Illumina MiSeq paired-end sequencing with 150 bp read length and cluster generation with 10% PhiX control DNA on an Illumina MiSeq platform (Illumina Inc., San Diego, CA).

Once sequencing was complete, corresponding paired-end reads were stitched to obtain an ultimate amplicon size of 255 bp. The sequencing data for each sample were screened for quality and paired-end reads were combined using mothur [v.1.40.4; 24]. Sequences that contained ambiguous bases, were shorter than 250bp, longer than 255bp, contained homopolymers > 8 bases in size, and potential chimeric sequences were removed. Remaining sequences were then clustered into operational taxonomic units (OTU) with a 99% sequence similarity based on a distance matrix generated in mothur. Consensus taxonomy for OTUs were assigned using the SILVA SSU database [version 132, 25]. Diversity indices, Shannon, Simpson, and Chao1, were computed as previously described [26].

The hidden-state predictions of gene families and their abundance within a given location (mucosa or digesta) were made using Phylogenetic Investigation of Communities by Reconstruction of Unobserved States 2 [PICRUSt2 v. 2.3.0b, 27], according to default parameters. Briefly, representative 16S rRNA amplicon sequences for each OTU were aligned and placed into a phylogenetic tree with relation to reference genomes from the Integrated Microbial Genomes database using HMMER [28], EPA-ng [29], and GAPPA [30]. Hidden-state gene predictions were determined by castor references [31] based on nearest-sequenced taxon index, with a threshold of 2 (OTUs with the nearest-sequenced taxon greater than 2 were removed). The hidden-state predictions of 2,064 enzymes from the Enzyme Commission (EC) database [32] were characterized. A total of 25 enzymes associated with arabinoxylan degradation, pentose metabolism, or short chain fatty acid (SCFA) production were selected (Table 1), normalized by predicted 16S rRNA gene copy number per OTU, and subsequentially analyzed using preplanned contrast.

## Statistics

Data were analyzed according to the following statistical model:

$$Y_{ijkl} = \mu + \tau_i + \upsilon_j + \rho_k + e_{ijkl}$$

Where $Y_{ijkl}$ is the observed value for l$^{th}$ experimental unit within the i$^{th}$ level of dietary treatment of the j$^{th}$ block for the l$^{th}$ pig in the k$^{th}$ replicate; $\mu$ is the general mean; $\tau_i$ is the fixed effect of the i$^{th}$ diet (i = 1 to 4); $\upsilon_j$ is the random effect of the j$^{th}$ block (j = 1 to 5); $\rho_k$ is the random effect of the k$^{th}$ replicate (k = 1 to 3); and $e_{ijkl}$ is the associated variance as described by the model for $Y_{ijkl}$ (l = 1 through 60).

The absolute abundance of the 100 most abundant OTUs among samples, relative abundance of classified genera, and hidden-state predictions of the 25 enzymes of interest were analyzed using a negative binomial distribution in GLIMMIX procedure of SAS (Version 9.4, SAS Inst., Cary, NC), and they were offset by the total library count for a given sample. All P-values were corrected for false discovery rates using the MULTITEST procedure of SAS. Diversity indices and were analyzed using PROC MIXED procedure of SAS. Least square means were separated using Fisher's Least Significant Difference test, and treatment differences were considered significant if P or Q values were < 0.05 and trends if 0.05 ≥ P or Q< 0.10. For the top 100 OTUs and genera classifications with a treatment Q value of < 0.05, the log$_2$-fold change was calculated comparing LF vs HF, HF vs. HF+XY, and HF vs. HF+AX. Moreover, preplanned contrast comparing the hidden-state predictions for the 25 EC in Table 1 for HF vs.

**Table 1. Enzymes associated with arabinoxylan degradation, pentose metabolism, or short chain fatty acid production that were selected for gene prediction-based analysis using Phylogenetic Investigation of Communities by Reconstruction of Unobserved States.**

| Enzyme Commission Number | Accepted Name |
|---|---|
| 1.1.1.9 | D-xylulose reductase |
| 1.1.1.27 | L-lactate dehydrogenase |
| 1.1.1.28 | D-lactate dehydrogenase |
| 2.3.1.8 | Phosphate acetyltransferase |
| 2.7.2.1 | Acetate kinase |
| 2.7.2.7 | Butyrate kinase |
| 2.7.2.15 | Propionate kinase |
| 2.7.1.17 | Xylulose kinase |
| 2.8.3.1 | Propionate CoA-transferase |
| 2.8.3.8 | Acetate CoA-transferase |
| 2.8.3.9 | Butyrate—acetoacetate CoA-transferase |
| 2.8.3.18 | Succinyl-CoA:acetate CoA-transferase |
| 3.1.1.73 | Ferulic acid esterase |
| 3.2.1.8 | Endo-1,4-β-xylanase |
| 3.2.1.22 | Alpha-galactosidase |
| 3.2.1.37 | Xylan 1,4-β-xylosidase |
| 3.2.1.55 | Non-reducing end $\alpha$-L-arabinofuranosidase |
| 3.2.1.136 | Glucuronoarabinoxylan endo-1,4-β-xylanase |
| 3.2.1.139 | Alpha-glucosiduronase |
| 3.2.1.156 | Oligosaccharide reducing-end xylanase |
| 4.1.2.9 | Phosphoketolase |
| 5.3.1.5 | Xylose isomerase |
| 6.2.1.1 | Acetate—CoA ligase |
| 6.2.1.5 | Succinyl-CoA synthetase (ADP-forming) |
| 6.2.1.13 | Acetate—CoA ligase (ADP-forming) |

HF+XY, and HF vs. HF+AX was analyzed utilizing a negative binomial distribution in GLIM-MIX procedure of SAS. The $\log_2$-fold change was calculated comparing HF vs. HF+XY, and HF vs. HF+AX for EC hidden-state predictions if $Q$ values were < 0.10.

## Results

### General microbiota characterizations

A total of 2,413,572 high-quality reads were obtained from ileal digesta samples after quality control, with a median of 40,958 sequences per sample. Subsequently, those sequences were classified into 1,538 OTUs (> 10 sequences per OTU), and taxonomically classified into 230 microbial genera. Characterization of phyla (Fig 1A) present in ileal digesta revealed microbiota stemming from the *Firmicutes* phylum dominated (87.22%), followed by smaller contributions from *Actinobacteria* (8.15%) and *Proteobacteria* (3.63%). When further characterized into families (Fig 1C), *Lactobacillaceae* (19.02%), *Clostridiaceae_1* (17.28%), *Erysipelotrichaceae* (13.83%), *Peptostreptococcaceae* (12.91%), and *Streptococcaceae* (12.73%) were the 5 most abundant families. Microbial genera that accounted for >1% of all classified OTUs are depicted in Fig 1E, and *Lactobacillus* (19%), *Clostridium_sensu_stricto_1* (16.96%), *Streptococcus* (12.72%), *Turicibacter* (7.32%), and *Terrisporobacter* (6.9%) were the 5 most abundant in ileal digesta.

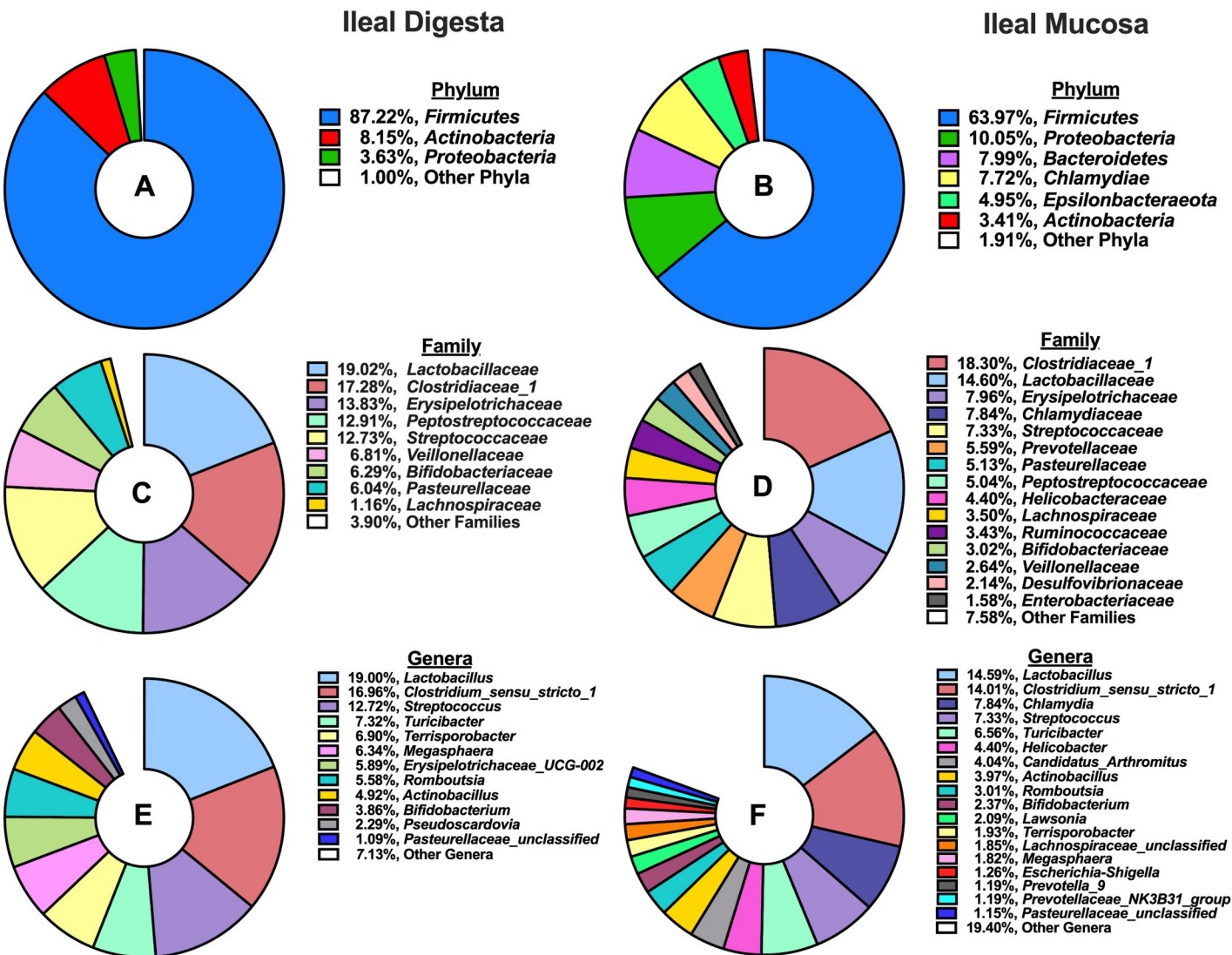

**Fig 1.** Summary of phylum, family, and genus-level microbial composition (>1% relative abundance) in the ileal digesta and mucosa among pigs (N = 60): A) Relative abundance of phyla present in the ileal digesta; B) Relative abundance of phyla present in the ileal mucosa; C) Relative abundance of families present in the ileal digesta; D) Relative abundance of families present in the ileal mucosa; E) Relative abundance of genera present in the ileal digesta; F) Relative abundance of genera present in the ileal mucosa.

From ileal mucosa samples, a total of 1,739,013 high-quality reads were obtained after quality control, with a median of 27,762 sequences per sample. A total of 2,495 OTUs (> 10 sequences per OTU) were classified and grouped into 317 genera. The 5 most abundant phyla, *Firmicutes* (63.97%), *Proteobacteria* (10.05%), *Bacteroidetes* (7.99%), *Chlamydia* (7.72%) and *Epsilonbactereota* (4.95%) accounted for 94.8% of all classified OTUs (Fig 1B). When further characterized into families (Fig 1D), *Clostridiaceae_1* (18.2%), *Lactobacillaceae* (14.6%), *Erysipelotrichaceae* (7.96%), *Chlamydiaceae* (7.84%), and *Streptococcaceae* (7.33%) were the 5 most abundant in the ileal mucosa. At a genus level, 15 microbial genera accounted for >1% of all classified OTUs, with *Lactobacillus* (14.59%), *Clostridium_sensu_stricto_1* (14.01%), *Chlamydia* (7.84%) *Streptococcus* (7.33%), and *Turicibacter* (6.56%) composing the top 5.

Shannon, Simpson, and Chao1 measures of diversity, evenness and richness are depicted in Fig 2. Numerically, ileal mucosa had a greater Shannon index relative to digesta (3.15 vs. 2.44),

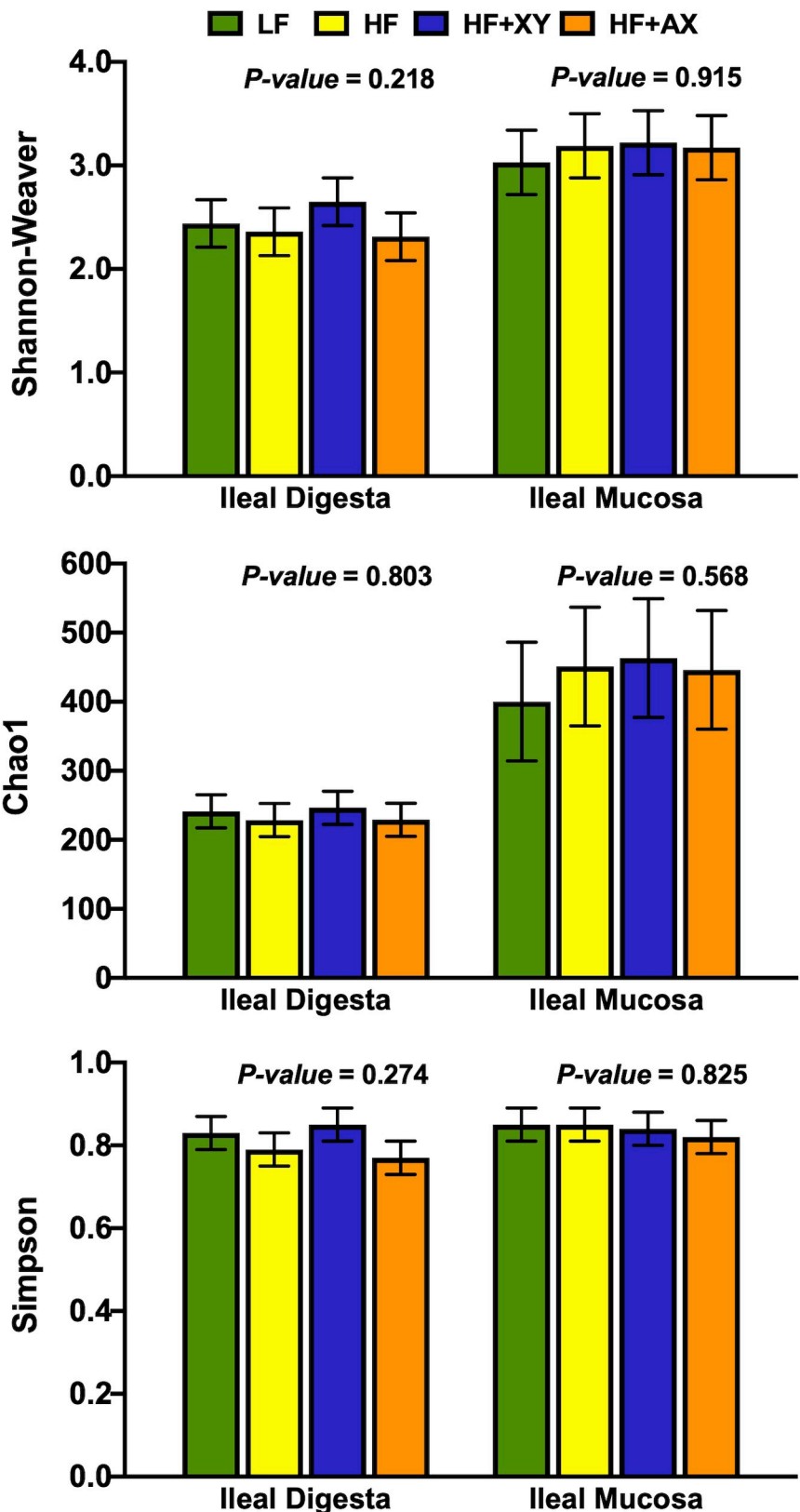

**Fig 2. Impact of treatment on measures of $\alpha$–diversity within the ileal digesta and mucosa.**

and similarly, the Chao1 index was greater for ileal digesta (440 vs. 236). However, the Shannon, Simpson, and Chao1 indices did not differ among treatments for both ileal digesta and mucosa ($P > 0.22$).

## Treatment modulation of microbial genera

Fig 3 represents the relative abundance of the 15 most abundant genera among treatments present in the ileal digesta (Fig 3A) and mucosa (Fig 3B). The top 15 genera account for 93.7%, 91.9%, 95.5%, and 93.9% of the OTUs characterized in the ileal digesta from pigs fed LF, HF,

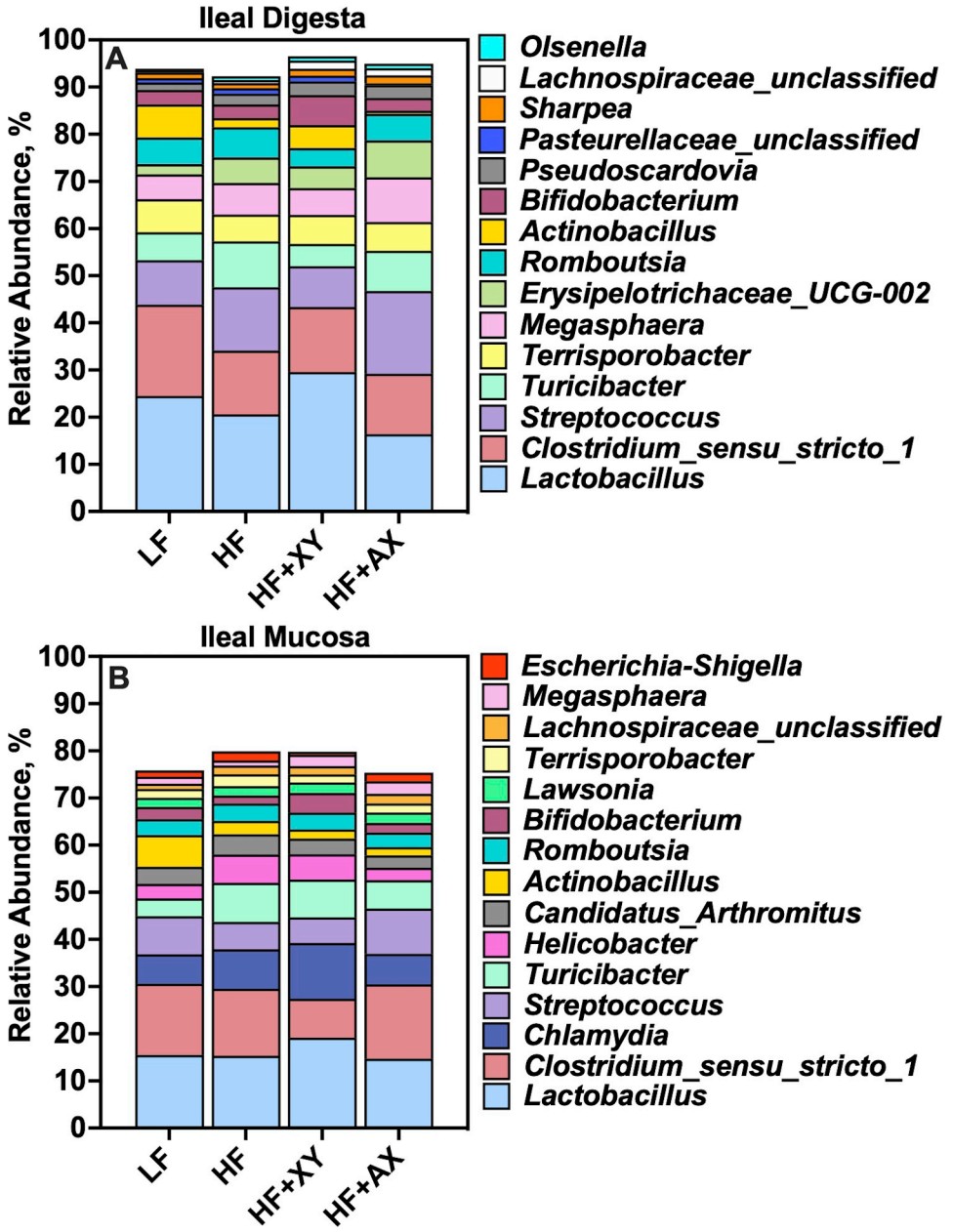

**Fig 3. The relative abundance of the 15 most abundant genera among treatments present in the ileal digesta and mucosa.**

HF+XY, and HF+AX, respectively. Likewise, the top 15 genera among microbiota characterized from ileal mucosa accounted for 75.1%, 79.0%, 79.1%, and 74.9% of the OTUs characterized in pigs fed LF, HF, HF+XY, and HF+AX, respectively.

Relative to LF, HF increased the abundance of *Erysipelotrichaceae_UCG-002*, *Olsenella*, and *Turicibacter*, and reduced *Actinobacillus* presences in ileal digesta (Fig 4; $Q < 0.05$). Likewise, compared to LF, HF increased the absolute abundance of *Turicibacter*, *Helicobacter*, *Lachnospiraceae_unclassified*, and decreased *Actinobacillus* presences in the ileal mucosa ($Q < 0.05$). The addition of xylanase to HF increased the prevalence of *Lachnospiraceae_unclassified*. *Actinobacillus*, *Bifidobacterium*, and *Lactobacillus*, and decreased the presence of *Streptococcus* and *Turicibacter* in ileal digesta ($Q < 0.05$). Moreover, relative to HF, HF+XY increased *Bifidobacterium*, *Megasphaera*, and *Chlamydia*, while reducing the presence of *Clostridium_sensu_stricto_1* and *Escherichia-Shigella* in the ileal mucosa ($Q < 0.05$). The direct supplementation of AXOS to HF increased the prevalence of *Lachnospiraceae_unclassified*, and decreased *Actinobacillus* and *Pasteurellaceae_unclassified* in the ileal digesta ($Q < 0.05$). Whereas, in the ileal mucosa, the addition of AXOS to HF increased the prevalence of *Megasphaera* and *Streptococcus*, but reduced *Candidatus_arthromitus* and *Helicobacter* ($Q < 0.05$).

## Treatment modulation of the 100 most abundant OTUs

Relative to LF, HF increased the prevalence of 9 of the 100 most abundant OTUs characterized from ileal digesta: OTU_95_*Megasphaera*, OTU_94_*Chlamydia*, OTU_12_*Terrisporobacter*, OTU_55_*Mitsuokella*, OTU_86_*Helicobacter*, OTU_62_*Actinobacillus*, OTU_36_*Olsenella*, OTU_25_*Sharpea*, and OTU_76_*Proteus* (Fig 5A; $Q < 0.05$). In contrast, compared to the LF, HF decreased 11 of the 100 most abundant OTUs found in ileal digesta: OTU_29_*Lactobacillus*, OTU_16_*Actinobacillus*, OTU_10_*Terrisporobacter*, OTU_27_*Lactobacillus*, OTU_18_*Actinobacillus*, OTU_23_*Actinobacillus*, OTU_67_*Lactobacillus*, OTU_7_*Streptococcus*, OTU_31_*Actinobacillus*, OTU_40_*Lachnospiraceae_unclassified* and, OTU_89_*Clostridium_sensu_stricto_1* (Fig 5A; $Q < 0.05$). Moreover, relative to LF, HF increased the abundance of 12 of the 100 most abundant OTUs characterized from ileal mucosa: OTU_14_*Helicobacter*, OTU_92_*Actinobacillus*, OTU_77_*Lachnospiraceae_unclassified*, OTU_93_*Lachnospiraceae_unclassified*, OTU_75_*Ruminococcaceae_UCG-014*, OTU_33_*Prevotellaceae_UCG-001*, OTU_65_*Lachnospiraceae_XPB1014_group*, OTU_36_*Ruminococcaceae_UCG-005*, OTU_44_*Gammaproteobacteria*, OTU_53_*Prevotellaceae_NK3B31_group*, OTU_39_*Terrisporobacter*, and OTU_51_*Lachnospiraceae_unclassified* (Fig 5B; $Q < 0.05$). Conversely, in relation to LF, HF decreased the prevalence of 10 of the 100 most abundant OTUs found in the ileal mucosa: OTU_83_*Pasteurellaceae_unclassified*, OTU_67_*Prevotella_9*, OTU_56_*Subdoligranulum*, OTU_80_*Agathobacter*, OTU_13_*Actinobacillus*, OTU_90_*Succinivibrio*, OTU_35_*Lactobacillus*, OTU_19_*Actinobacillus*, OTU_32_*Prevotella_9*, and OTU_12_*Megaspharea* (Fig 5B; $Q < 0.05$).

The addition of xylanase to HF (HF+XY), increased 19 of the 100 most abundant OTUs (Fig 6A; $Q < 0.05$). Notably, HF+XY increased the prevalence of 3 OTUs (OTU # 27, 29, and 38) stemming from the *Lactobacillus* genus, 3 OTUs (OTU # 61, 73, and 83) from the *Alloprevotella* genus, and 5 OTUs (OTU # 16, 18, 21, 23, 29, and 31) from the *Actinobacillus* genus in ileal digesta ($Q < 0.05$). Interestingly, relative to HF, HF+XY increased the abundance of OTU_23_*Faecalibacterium*, OTU_40_*Lachnospiraceae_unclassified*, OTU_74_*Subdoligranulum*, OTU_82_*Streptococcus*, OTU_87_*Prevotellaceae_UCG-003*.

OTU_90_*Gammaproteobacteria_unclassified*, each by over 6 $\log_2$-fold ($Q < 0.05$). In contrast, the addition of xylanase to HF decreased 8 of the 100 most abundant OTUs, and notably, this included OTU_86_*Helicobacter*, OTU_94_*Chlamydia*, and OTU_95_*Megasphaera* (Fig

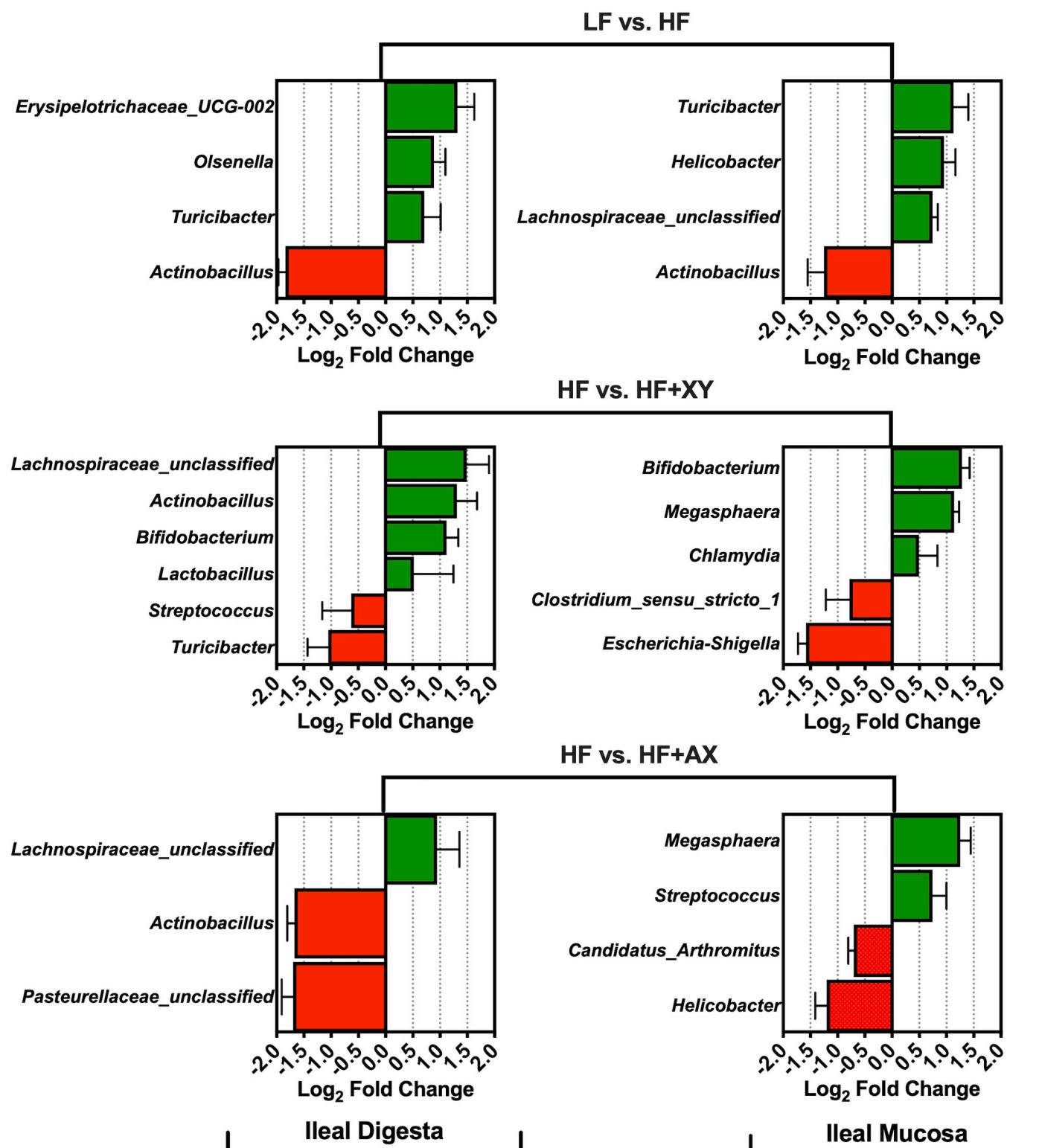

**Fig 4. The log$_2$-fold change difference between LF vs. HF, HF vs. HF+XY, HF vs. HF+AX for the 15 most abundant microbial genera present in the ileal digesta and mucosa with a Q-value < 0.05.**

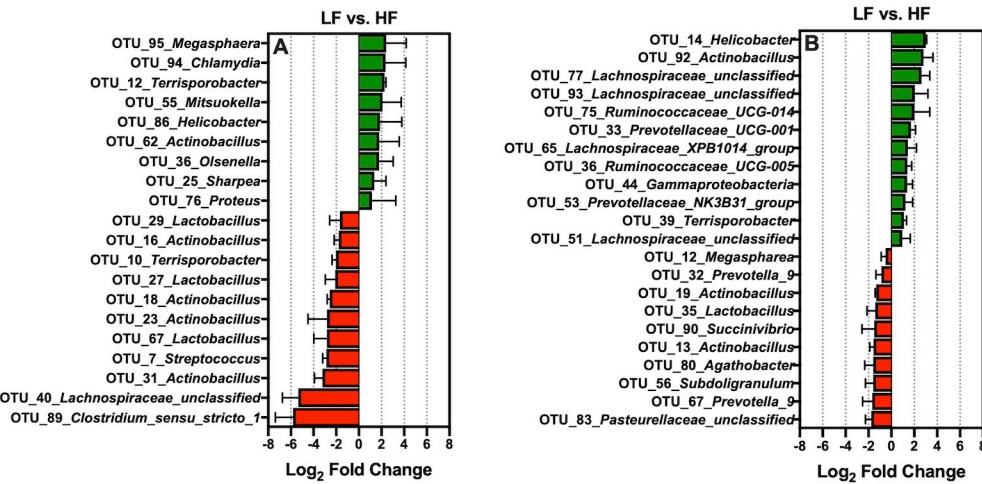

**Fig 5.** The log$_2$-fold change difference between LF vs. HF for the significant OTUs from the 100 most abundant OTUs among treatments present in the ileal digesta (A) and mucosa (B).

6A; $Q < 0.05$). With regards to the ileal mucosa, HF+XY increased the abundance of 14, and decreased 8 of the 100 most abundant OTUs (Fig 6B; $Q < 0.05$). Notably, in the ileal mucosa, HF+XY increased the prevalence of 3 OTUs (OTU # 34, 35, and 38) stemming from the *Lactobacillus* genus, and also increased OTU_16_*Bifidobacterium* by nearly 2 log$_2$-fold. On the other hand, pigs fed HF+XY had decreased prevalence of OTU_92_*Actinobacillus*, OTU_77_ *Lachnospiraceae_unclassified*, OTU_33_*Prevotellaceae_UCG-001*, OTU_36_*Ruminococcaceae-UCG-005*, OTU_44_*Gammaproteobacteria*, OTU_53_*Prevotellaceae_NK3B31_group*, OTU_87_*Prevotellaceae_unclassified*, and OTU_51_*Lachnospiraceae_unclassified*, in the ileal mucosa (Fig 6B; $Q < 0.05$).

The addition of AXOS to HF (HF+AX), increased 5 of the 100 most abundant OTUs found in ileal digesta: OTU_39_*Sarcina*, OTU_40_*Lachnospiraceae_unclassified*, OTU_7_*Streptococcus*, OTU_35_*Enterobacteriaceae_unclassified*, and OTU_56_*Lactobacillus* (Fig 7A; $Q < 0.05$).

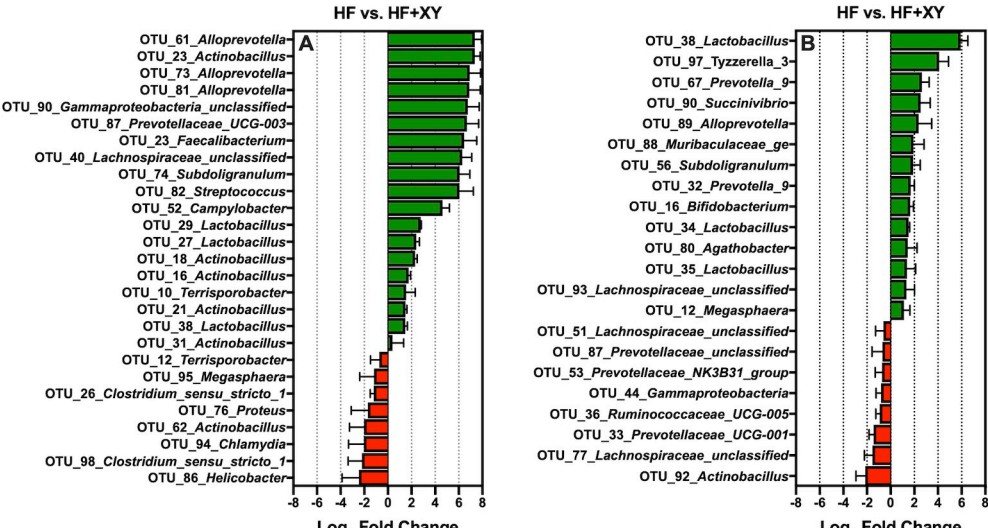

**Fig 6.** The log$_2$-fold change differences between HF vs. HF+XY for the significant OTUs from the 100 most abundant OTUs among treatments present in the ileal digesta (A) and mucosa (B).

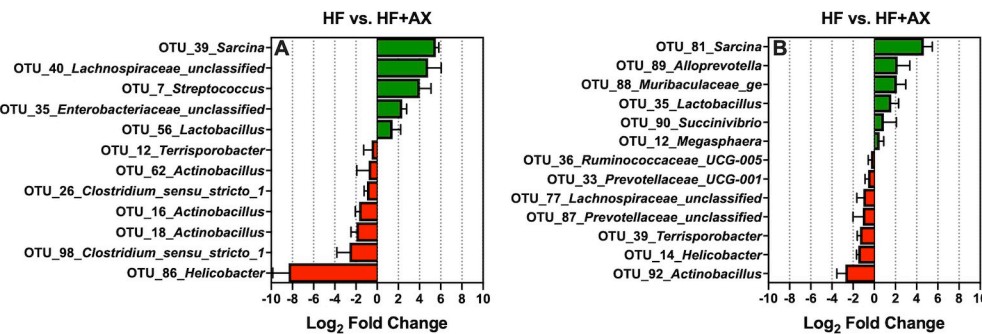

**Fig 7.** The $\log_2$-fold change differences between HF vs. HF+AX for the significant OTUs from the 100 most abundant OTUs among treatments present in the ileal digesta (A) and mucosa (B).

In contrast, relative to HF, HF+AX decreased 7 of the 100 most abundant OTUs found in ileal digesta: OTU_86_*Helicobacter*, OTU_26 and OTU_98 (both:*Clostridium_sensu_stricto_1*), OTU_16, OTU_18, and OTU_62 (all: *Actinobacillus*), and OTU_12_*Terrisporobacter* (Fig 7A; $Q < 0.05$). For the mucosa, relative to HF, HF+AX increased the abundance of 6 of the 100 most abundant OTUs: OTU_81_*Sarcina*, OTU_89_*Alloprevotella*, OTU_88_*Muribaculaceae_ge*, OTU_35_*Lactobacillus*, OTU_90_*Succinivibrio*, and OTU_12_*Megasphaera* (Fig 7B; $Q < 0.05$). Additionally, relative to HF, HF+AX decreased the prevalence of 7 of the 100 most abundant OTUs: OTU_92_*Actinobacillus*, OTU_14_*Helicobacter*, OTU_39_*Terrisporobacter*, OTU_87_*Prevotellaceae_unclassified*, OTU_77_*Lachnospiraceae_unclassified*, OTU_33_*Prevotellaceae_UCG-001*, and OTU_36_*Ruminococcaceae_UCG-005* (Fig 7B; $Q < 0.05$).

## PICRUST2 targeted gene predictions

In the ileal digesta, relative to HF, HF+XY tended to upregulate butyrate kinase and non-reducing end $\alpha$-L-arabinofuranosidase (Fig 8; $Q < 0.10$). Moreover, the addition of xylanase to HF, increased the predicted gene counts for endo-1,4-β-xylanase, glucuronoarabinoxylan endo-1,4-β-xylanase, oligosaccharide reducing-end xylanase, xylan 1,4-β-xylosidase, xylose isomerase, and xylulose kinase in the ileal digesta (Fig 8; $Q < 0.05$). Relative to HF, HF+AX increased oligosaccharide reducing-end xylanase and xylan 1,4-β-xylosidase (Fig 9; $Q < 0.10$) and tended to increase xylose isomerase (Fig 8; $Q < 0.05$).

In the ileal mucosa, HF+AX did not differ from HF for any of the chosen enzymes in Table 1 ($Q > 0.10$). However, compared to HF, HF+XY increased the predicted gene counts for acetate CoA-transferase, acetate kinase, acetate—CoA ligase, alpha-galactosidase, butyrate kinase, butyrate—acetoacetate CoA-transferase, ferulic acid esterase, oligosaccharide reducing-end xylanase, succinyl-CoA synthetase, succinyl-CoA:acetate CoA-transferase, and xylan 1,4-β-xylosidase in the ileal mucosa (Fig 9; $Q < 0.05$).

## Discussion

In a healthy state, the pig and gastrointestinal microbiota share a symbiotic relationship that contributes to the nutrition and health of the pig, and in turn, the pig provides substrate for microbiota largely in the form of DF. However, all DF is not created equal and its efficacy to modulate microbial taxa, diversity, and composition is dependent on DF composition, solubility, and concentration [33,34]. In general, soluble DF and non-digestible oligosaccharides are fermented at a faster rate compared to insoluble DF [35,36] and are associated with reducing pathogenetic microbes, increasing microbial diversity, and improving the proliferation of

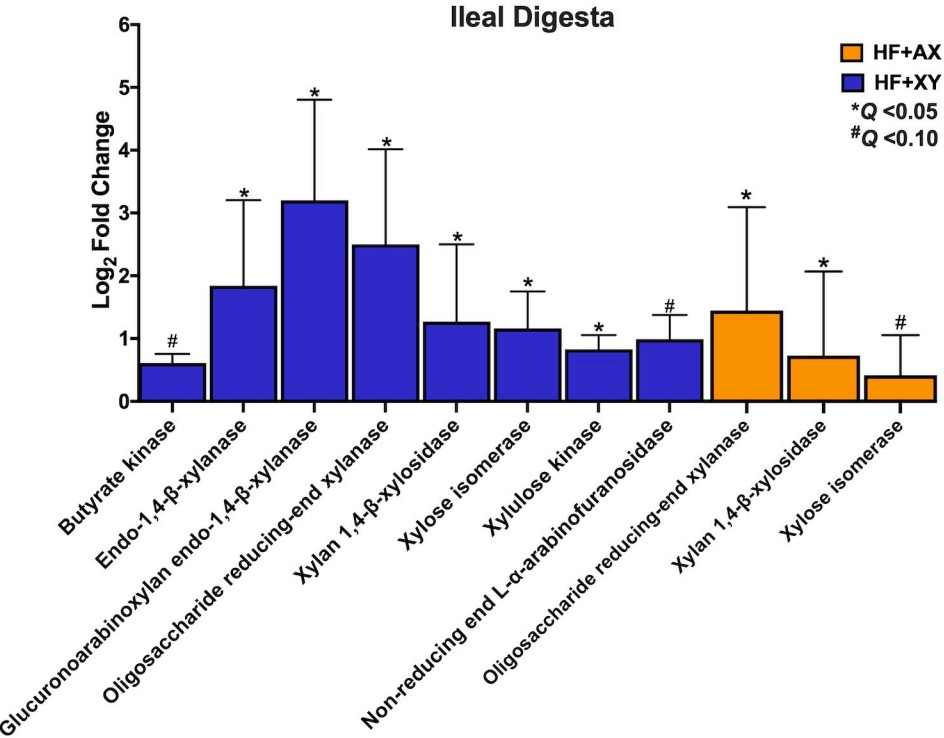

**Fig 8. The log$_2$-fold change between HF vs. HF+XY or HF vs. HF+AX for the PICRUST2 predicted gene counts for enzymes associated with bacterial arabinoxylan degradation, xylose metabolism, or short chain fatty acid metabolism in the ileal digesta that tended or significantly differed.**

'beneficial' microbes [37,38]. In contrast, swine diets are largely composed of poorly fermentable insoluble DF emanating from cereal grains and industrial co-products [8,39]. Globally, corn is the most abundant cereal grain, and thus, swine diets often contain considerable amounts of insoluble corn-based fiber, particularly arabinoxylans, that are poorly fermented by the pig's gastrointestinal microbiota [3,5]. Xylanase, a carbohydrase that targets arabinoxylans, may modulate the gastrointestinal microbial ecology of pigs fed corn-based diets by hydrolyzing arabinoxylans into smaller and more fermentable fragments [i.e. AXOS; 15]. Indeed, it is often postulated that the unexpected health benefits observed with xylanase supplementation are a result of enteric microbiota modulation [16,17]. Still, the impact of xylanase, particularly its mode of action, on gastrointestinal microbiota in pigs fed corn-based fiber is largely unknown. Thus, the experimental objective was to characterize the impact of increased insoluble fiber, xylanase, and directly supplemented AXOS on the ileal digesta and mucosa microbiome of pigs fed corn-based fiber.

The composition of microbial phyla summarized in Fig 1 and is comparable with other reports of ileal digesta and mucosa microbiota composition in pigs under commercial conditions [40,41]. Similar to herein, a meta-analysis of 16S rRNA bacterial gene sequences across the gastrointestinal tract of the pig identified *Clostridium* and *Lactobacillus* as the most prevalent genera in the ileal digesta and mucosa [40]. However, there were some differences among the most abundant genera, compared to Holman et al., [40]; this is likely driven by differences in diet, host genetics, and relative environment. Moreover, Zhang et al., [42] also observed numerically higher OTU counts, Shannon index values, and Chao1 index values in the ileal mucosa relative to ileal digesta in pigs fed a corn-based diet (Fig 2). The increased richness and diversity of mucosal-associated microbiota is not unanticipated as we observed a greater

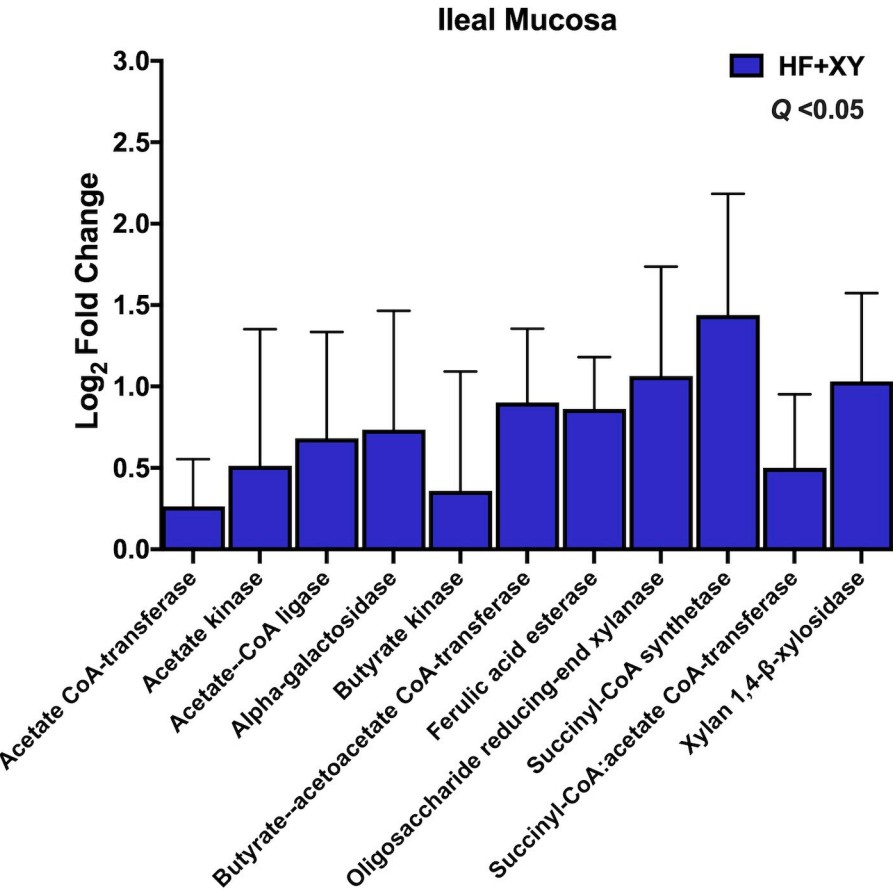

**Fig 9. The log$_2$-fold change between HF vs. HF+XY for the PICRUST2 predicted gene counts for enzymes associated with bacterial arabinoxylan degradation, xylose metabolism, or short chain fatty acid metabolism in the ileal mucosa that tended or significantly differed.**

sequence count and OTU abundance in the mucosa, relative to the digesta. These differences in microbial communities are probably a result of differing nutrient availability and solubility, associated commensal microbiota, and environmental differences, particularly the presence of oxygen found in the epithelial mucosa versus the lumen of the ileum [43].

When comparing HF to LF, HF increased the abundance of *Erysipelotrichaceae_UCG-002*, *Olsenella*, and *Turicibacter* in the ileal digesta (Fig 4A). While the role of genera stemming from *Erysipelotrichaceae* is largely unknown in the pig, in humans they are associated with subjects who consume high-fat western diets and may increase gut inflammation through immunogenic properties [44]. However, the acid ether-extract content of LF and HF were relatively similar [14], and certainly, the fiber level in HF is not typical of a human western diet. Potentially, the increased endogenous fat losses associated with diets high in insoluble fiber favored the metabolism of *Erysipelotrichaceae* [45], but this is purely speculative. Moreover, an increase of OTUs stemming from *Prevotella*, *Ruminococcaceae*, and *Lachnospiraceae* observed in Fig 5B suggest a greater abundance of mucins and sloughed cells in HF as these microbiota digest and metabolize mucins and sloughed epithelial cells through cooperative metabolism [46,47]. An increase in *Turicibacter* was observed in both mucosa and digesta of pigs fed HF, relative to LF; this is in agreement with Le Sciellour [48] who observed a greater abundance of *Turicibacter* in feces from pigs fed a diet with 20% NDF compared to a diet with 10% NDF. However, an increased abundance of *Turicibacter* in feces has been positively correlated with

increased body weight in grow-finish pigs [49], but this is counterintuitive to what was observed in a related study, as pigs fed HF weighed significantly less than LF [14]. Pigs fed HF had a lower abundance of *Actinobacillus* present in both ileal digesta and mucosa (Fig 4A and 4B). Little is known about the *Actinobacillus* genus with the exception of species known to cause oral and respiratory diseases [50]. It is unclear if *Actinobacillus* stemming from the gastrointestinal tract are disease causing or commensal. Notably, work by Gottschalk et al., [51] discovered two non-pathogenic *Actinobacillus* species in healthy pigs that were antigenically and biochemically similar to *A. pleuropneumoniae;* research characterizing ileal associated *Actinobacillus* in the pig is thus warranted.

In this study, the addition of xylanase increased the prevalence of several genera, and OTUs, generally recognized as beneficial, *Bifidobacterium*, *Lactobacillus*, *Lachnospiraceae_unclassified*, *Faecalibacterium*, and *Alloprevotella*, and decreased two genera known to contain potentially disease-causing species, *Streptococcus* and *Escherichia-Shigella* (Figs 4 and 6). The upregulation of *Bifidobacterium*, *Lachnospiraceae_unclassified*, and *Lactobacillus* by xylanase is in agreement with xylanase supplementation in poultry and turkey diets [52,53]. However, the modulation by xylanase in this study is in contrast to work from Zhang et al., [54] who evaluated the ileal microbiota from pigs fed five mono-component xylanases in corn distiller's dried grain with solubles (DDGS)-based diets. This is potentially a result of utilizing corn-bran without solubles herein as the main source of corn-based fiber instead of DDGS as corn-bran appears to be more susceptible to xylanase hydrolysis [14]. Indeed, related work from the same animal study observed HF+XY increased the apparent ileal digestibility of insoluble arabinoxylans by 54% compared to HF [55].

The increased abundance of *Bifidobacterium*, *Lachnospiraceae_unclassified*, *Lactobacillus*, and *Faecalibacterium* suggest xylanase promotes a symbiotic cooperation for cross-feeding of the potential release products of xylanase that likely increases SCFA production, particularly acetate and butyrate; this cooperation would coincide with a prebiotic-like mechanism [56]. Moreover, the increase in predicted gene counts of enzymes associated with AXOS degradation and SCFA production also support this hypothesis (Figs 8 and 9). Theoretically, the AXOS released by xylanase could be metabolized by *Lactobacillus*, *Bifidobacterium*, and *Lachnospiraceae_unclassified* associated microbiota as several strains from these genera produce many of the accessory enzymes needed to hydrolyze xylo-oligosaccharides and AXOS [57–59]. However, depending on the degrees of polymerization of the AXOS produced by xylanase, these genera likely utilize AXOS with varying efficiencies [60]. *Bifidobacterium* preferentially favor the uptake of short AXOS into the cell, where they are further degraded to arabinose and xylose; these pentoses are shuttled into the bifid shunt by their conversion into xylulose 5-phosphate, and ultimately the production of acetate [61]. Indeed, xylanase supplementation favored ileal digesta microbiota with a higher predicted gene count for xylose isomerase and xylulose kinase (Fig 8), and acetate kinase producing bacteria in the ileal mucosa (Fig 9). Several strains of *Lactobacillus* are known to degrade higher molecular weight AXOS [58,59], and there is genomic and *in vitro* evidence to suggest species deriving from *Lachnospiraceae* can hydrolyze arabinoxylan and AXOS, and metabolize xylose [62,63]. Finally, the upregulation of *Faecalibacterium* by HF+XY, whose solely known species is *Faecalibacterium prausnitzii*, is most likely an indirect result of cross-feeding on the acetate produced by *Lactobacillus* and *Bifidobacterium* as *Faecalibacterium* cannot metabolize pentose monomers [64]. This is further supported by the upregulation of butyryl-CoA:acetate CoA transferase and acetate kinase predicted genes by HF+XY, two key enzymes in the conversion of acetate to butyrate by *Faecalibacterium prausnitzii*, [65,66]. Thus, the increased modulation of *Bifidobacterium*, *Lachnospiraceae_unclassified*, *Lactobacillus*, and *Faecalibacterium* by HF+XY suggest there is likely a direct upregulation of acetate and butyrate in the ileum via AXOS and acetate metabolism, respectively.

It is well established that increased butyrate will enhance gut barrier integrity through mediation of the nuclear NF-kB pathway and histone deacetylase inhibition, aid in the control of enteric pathogens, and reduce inflammation and oxidative stress [67,68]. Similarly, acetate can modulate intestinal epithelial integrity and homeostasis via G protein coupled FFAR2 receptors that activate the NLRP3 inflammasome pathway [69]. Indeed, research by Petry et al. [14] from the same animal study discovered that HF+XY increased total antioxidant capacity of the ileum, reduced serum malondialdehyde concentrations, and increased ileal mRNA of claudin 4 and occludin tight junction proteins. Interestingly, xylanase also increased predicted gene counts for ferulic acid esterase; ferulic acid is a potent antioxidant capable of scavenging free radicals and inhibiting free radical production [70]. Corn arabinoxylan is highly substituted with ferulic acid, but its bioavailability is reduced due to esterification within arabinoxylan [71]. It is plausible that hydrolysis of arabinoxylan by HF+XY released feruloylated AXOS that could be further degraded by microbial ferulic acid esterase to improve ferulic acid bioavailability, and possible contribute to the total antioxidant capacity of the pig [72]. Indeed, supplementation of corn-based feruloylated oligosaccharides in rats increased *Lactobacillus* abundance, and decreased *Streptococcus* and *Clostridium*, which is similar to what was found in HF+XY herein [73].

Overall, the improved oxidative status and gut barrier integrity observed by Petry et al., [14], and the upregulation of beneficial acetate and butyrate producing bacteria observed herein, favors a prebiotic like mechanism. However, the increased modulation of predicted genes associated with arabinoxylan degradation in the ileal digesta favor the definition of a stimbiotic mechanism [16]. A holistic view of these data suggests both of these mechanisms are likely occurring synergistically in that xylanase acts as a stimbiotic in the digesta increasing arabinoxylan solubilization and AXOS production, and then in turn, the released AXOS acts in a prebiotic-like manner with mucosal-associated bacteria. Further research is warranted to determine the composition of AXOS produced *in situ* by xylanase in the presence of corn-based fiber. However, due to the chemical complexities of corn-based arabinoxylans, determining the composition of *in situ* produced AXOS from corn-based arabinoxylans is an analytically challenging pursuit that has yet to be mastered [74].

Directly supplementing AXOS (HF+AX) did not modulate ileal microbiota to the same degree as xylanase (Fig 7); this is in contrast to what has been reported in poultry [75,76]. Potentially, supplementing xylanase directly may have produced a greater ileal AXOS concentration than what was supplemented in HF+AX, and increasing the concentration within the diet may improve its efficacy. Moreover, the *in situ* produced AXOS from corn-based fiber may differ in composition to what was supplemented (i.e. DP and substitution pattern). It is also plausible that the directly supplemented AXOS may have been fermented prior to reaching the ileum of the pig. However, when compared to HF, HF+AX did increase the abundance of OTU_40_*Lachnospiraceae_unclassified*, and OTU_56_*Lactobacillus* and this is likely through fermentation of AXOS as previously described. This is also supported by the increased abundance of the predicted genes for oligosaccharide reducing-end xylanase, xylan 1,4-β-xylosidase, and xylose isomerase by HF+AX in the ileal digesta (Fig 8). Interestingly, HF+AX greatly increased OTU_39_*Sarcina* and OTU_81_*Sarcina*, relative to HF, in the digesta and mucosa, respectively. There is a dearth of information about *Sarcina* in the pig but has been associated with gastrointestinal disease and delayed gastric emptying in humans [77].

## Conclusions

Increasing insoluble corn-based DF through the addition of corn bran without solubles modulated microbial communities associated with mucin and epithelial cell degradation and fat

metabolism, while decreasing the abundance of *Actinobacillus* that is often associated with disease causing species. The direct supplementation of AXOS did not modulate ileal microbiota in a similar fashion or to same degree as xylanase but did favor bacteria with increased genes for oligosaccharide reducing-end xylanase, xylan 1,4-β-xylosidase, and xylose isomerase in the digesta. Directly-supplemented AXOS may differ in composition from what is produced *in situ* by xylanase's degradation of corn-based arabinoxylan, or xylanase may increase AXOS production to a greater concentration than what was supplemented in HF+AX. However, supplementing xylanase to HF modulated ileal microbiota to favor beneficial microbial communities that are associated with the symbiotic cross-feeding of oligosaccharides derived from AXOS. The addition of xylanase also upregulated predicted gene counts for enzymes associated with arabinoxylan and AXOS degradation, pentose metabolism, and SCFA production. These data, in combination with the unexpected health benefits often reported in the literature, suggest xylanase favors a prebiotic-like mechanism likely through the release of AXOS that modulate the cross-feeding metabolism of certain beneficial microbial communities. However, the increased predicted gene counts for enzymes associated with arabinoxylan and AXOS degradation in the ileal digesta favors a stimbiotic mechanism. Potentially, xylanase could act as a stimbiotic in the digesta increasing arabinoxylan solubilization and AXOS production, and then in turn, the released AXOS act in a prebiotic-like manner with mucosal associated bacteria.

## Acknowledgments

The authors would like to thank Iowa Corn Processors, DSM and Ajinomoto Heartland, Inc for providing in-kind contributions to the Applied Swine Nutrition program at Iowa State University.

## Author Contributions

**Conceptualization:** Amy L. Petry, John F. Patience, Nichole F. Huntley, Michael R. Bedford.

**Data curation:** Amy L. Petry.

**Formal analysis:** Amy L. Petry, Lucas R. Koester.

**Funding acquisition:** John F. Patience, Nichole F. Huntley, Michael R. Bedford.

**Investigation:** Amy L. Petry.

**Methodology:** Amy L. Petry, John F. Patience, Lucas R. Koester, Nichole F. Huntley, Michael R. Bedford, Stephan Schmitz-Esser.

**Project administration:** John F. Patience.

**Resources:** John F. Patience.

**Supervision:** John F. Patience, Stephan Schmitz-Esser.

**Validation:** Amy L. Petry.

**Visualization:** Amy L. Petry.

**Writing – original draft:** Amy L. Petry.

**Writing – review & editing:** Amy L. Petry, John F. Patience, Lucas R. Koester, Nichole F. Huntley, Michael R. Bedford, Stephan Schmitz-Esser.

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
