## [Decision Letter · Decision Letter 0]

6 Nov 2020

PONE-D-20-30575

Xylanase modulates the microbiota of ileal mucosa and digesta of pigs fed corn-based arabinoxylans likely through both a stimbiotic and prebiotic mechanism

PLOS ONE

Dear Dr. Schmitz-Esser,

Thank you for submitting your manuscript to PLOS ONE. After careful consideration, we feel that it has merit but does not fully meet PLOS ONE’s publication criteria as it currently stands. Therefore, we invite you to submit a revised version of the manuscript that addresses all the points raised during the review process.

Notably, you will see that both reviewers asked for additional information regarding your study.

We look forward to receiving your revised manuscript.

Kind regards,

F. Blachier, PhD

Academic Editor

PLOS ONE

Journal Requirements:

2. We note that you are reporting an analysis of a microarray, next-generation sequencing, or deep sequencing data set. PLOS requires that authors comply with field-specific standards for preparation, recording, and deposition of data in repositories appropriate to their field. Please upload these data to a stable, public repository (such as ArrayExpress, Gene Expression Omnibus (GEO), DNA Data Bank of Japan (DDBJ), NCBI GenBank, NCBI Sequence Read Archive, or EMBL Nucleotide Sequence Database (ENA)). In your revised cover letter, please provide the relevant accession numbers that may be used to access these data. For a full list of recommended repositories, see http://journals.plos.org/plosone/s/data-availability#loc-omics or http://journals.plos.org/plosone/s/data-availability#loc-sequencing.

"Financial support for this research was provided by AB Vista Feed Ingredients and by the Agriculture and Food Research Initiative Competitive Grant nos. 2016-38420-25496 from the United States Department of Agriculture National Institute of Food and Agriculture.

We note that one or more of the authors have an affiliation to the commercial funders of this research study : [AB Vista Feed Ingredients].

3.1. Please provide an amended Funding Statement declaring this commercial affiliation, as well as a statement regarding the Role of Funders in your study. If the funding organization did not play a role in the study design, data collection and analysis, decision to publish, or preparation of the manuscript and only provided financial support in the form of authors' salaries and/or research materials, please review your statements relating to the author contributions, and ensure you have specifically and accurately indicated the role(s) that these authors had in your study. You can update author roles in the Author Contributions section of the online submission form.

3.2. Please also provide an updated Competing Interests Statement declaring this commercial affiliation along with any other relevant declarations relating to employment, consultancy, patents, products in development, or marketed products, etc.  

Reviewers' comments:

Reviewer's Responses to Questions

**Comments to the Author**

1. Is the manuscript technically sound, and do the data support the conclusions?

Reviewer #1: Yes

Reviewer #2: Yes

2. Has the statistical analysis been performed appropriately and rigorously? 

Reviewer #1: Yes

Reviewer #2: Yes

3. Have the authors made all data underlying the findings in their manuscript fully available?

Reviewer #1: Yes

Reviewer #2: Yes

4. Is the manuscript presented in an intelligible fashion and written in standard English?

Reviewer #1: Yes

Reviewer #2: Yes

5. Review Comments to the Author

Reviewer #1: In this study, Petry et al. investigated the effects of supplementation of xylanase to a high fiber diet on the microbial diversities of ileal mucosa and digesta of pigs, overall the design of this study is elegant and smart, and the findings are interesting, and I would recommend publication in Plos one. I only have some minor concerns.

1. The specific nature and effectiveness of xylanase used in this study. The use of xylanase to a high fiber diet in this study brings smartness in the design of this study, thus it would be important to see the nature and effectiveness of xylanase. Could this study provide some evidence that the arabinoxylans was degraded by xylanase to produce smaller fermentable fragments like AXOS?

2. Why only microbiota in the ileal digesta or mucosa were determined, while colonic microbiota was dominated along the whole gut?

3. The arabinoxylan-oligosaccharide addition to the HF diet did not always produce similar effects to that of xylanase group, that might be due to the level of supplementation. In this study, 50 mg arabinoxylan-oligosaccharide/kg were added in the HF diet, could this level elicit similar effects of xylanase?

4. In deed, this is a related study of Petry et al. (Journal of Animal Science, 2020, Vol. 98, No. 7, 1–11, doi:10.1093/jas/skaa233), if it is possible, some of the data in J Anim Sci (i.e., antioxidant parameter)could be used to establish relationship with the microbial data in this study.

Reviewer #2: Petry et al. had studied “Xylanase modulates the microbiota of ileal mucosa and digesta of pigs fed corn-based arabinoxylans likely through both a stimbiotic and prebiotic mechanism”. They conclude that the addition of xylanase or AXOS would modulate intestinal microbial communities that support fiber degradation (stimbiotic mechanism) or would be selectively utilized by communities that are known to confer a health benefit (prebiotic mechanism). The topic is interesting and the research gives a new insight on relationships between microbial community and xylanase/AX/high fibers derived from wheat bran. However, many points should be solved and explained before publication.

Major comments

1. Experimental design. A total of 60 growing gilts (progeny of Camborough sows × 337 sires; PIC Inc., Hendersonville, TN) were used in 3 replicates (20 gilts per replicate) of a 46-day trial. Gilts were individually housed in pens for 36 days, and subsequently moved to metabolism crates for the previously published 10-d metabolism study. So how many replicates/pigs are per treatment/replicate? It is not clear to readers.

2. Materials and methods: where are AX and xylanase from? Is AX extracted from corn bran? And also xylanase. Please note that different sources of AX exert different effects of microbial community composition.

3. Results: In the previous study by Zhao et al. (2019), Firmicutes and Proteobacteria were the two dominant phyla in the ileum digesta, which is in agreement with microbial community of the ileal mucosa in the current study. (Zhao et al. Fiber-rich foods affected gut bacterial community and short-chain fatty acids production in pig model.) However, how to explain differences in microbial composition between ileal digesta and ileal mucosa.

4. Discussion: Xylanase plays positive actives in the foregut of pigs, such as decreasing viscosity of the intestinal digesta and increasing nutrient digestibility. However, AX, derived from corn and sorghum, is primarily fermented in the hindgut of pigs. Therefore, it is not reasonable to compare effects of xylanase and AX on microbial community in the ileum of the pig. Please discuss it.

5. Conclusion: The context in conclusion is redundant, and most of points are only conjectural. Such as Line 502-505, because no data for SCFA concentration was shown in the current study.

Specific comments

1. Line 29: 16S rRNA?

2. Line 49-53. Many recent study have reported prebiotic effects of dietary fiber on weanling pigs nutrition.

3. Line 353-360: This part described different responses of IDF and SDF on pig nutrition, but it is not relevance to objectives of the present study.

4. Line 364-372: those contexts should be moved to the part of Introduction.

6. PLOS authors have the option to publish the peer review history of their article (what does this mean?). If published, this will include your full peer review and any attached files.

Reviewer #1: No

Reviewer #2: No

---

## [Author Response · Author response to Decision Letter 0]

14 Dec 2020

Responses to reviewer’s comments on manuscript Xylanase modulates the microbiota of ileal mucosa and digesta of pigs fed corn-based arabinoxylans likely through both a stimbiotic and prebiotic mechanism

 (PONE-D-20-30575) 

Reviewer 1 Comments

General Comments. Revision: Line 436-437 

“ #1. The specific nature and effectiveness of xylanase used in this study. The use of xylanase to a high fiber diet in this study brings smartness in the design of this study, thus it would be important to see the nature and effectiveness of xylanase. Could this study provide some evidence that the arabinoxylans was degraded by xylanase to produce smaller fermentable fragments like AXOS?”

Response to the reviewer: Thank you for this comment. We agree that confirming in the minds of the reader that xylanase functioned as expected is helpful. A sentence regarding an improvement in arabinoxylan digestibility previously reported has been added to lines 436-437 in the discussion. Additionally, our lab is currently pursuing a semi quantitative oligomer release pattern of xylanase using HPLC and HPAEC-PAD for a separate forthcoming publication which will have different objectives. 

General Comments. No Revision 

“ #2. Why only microbiota in the ileal digesta or mucosa were determined, while colonic microbiota was dominated along the whole gut?”

Response to the reviewer: The ileum was the target location of this study due to the modulation of gastrointestinal barrier integrity and oxidative status by xylanase reported in Petry et al, 2020. These findings led to an investigation of the prebiotic effect of xylanase supplementation; and in order to fulfill the definition of a prebiotic, a health benefit must be conferred. Our investigation in the modulation of colonic health did not reveal a benefit for the markers evaluated. However, large intestine microbiota was investigated in a different publication (under review) in relation to fiber fermentation and not the prebiotic MOA. Since the focus of our study of the ileal microbiome is quite different from that of the colonic microbiome, there was no value in including the 2 studies in the same manuscript. 

General Comments. Revision: Line 513-514

“ #3 The arabinoxylan-oligosaccharide addition to the HF diet did not always produce similar effects to that of xylanase group, that might be due to the level of supplementation. In this study, 50 mg arabinoxylan-oligosaccharide/kg were added in the HF diet, could this level elicit similar effects of xylanase?”

Response to the reviewer: There is a dearth of literature on the optimal supplementation level of AXOS in swine. Research evaluating the optimal supplementation level of AXOS has been predominantly conducted in poultry or derived from the concentration of AXOS released under in vitro conditions. The supplementation level in this study was based on that body of literature. The authors agree that further research is needed to determine if a greater level of supplementation would improve its efficacy. This is a good point and has now been added to the discussion. 

General Comments. No Revision 

“ #4 Indeed, this is a related study of Petry et al. (Journal of Animal Science, 2020, Vol. 98, No. 7, 1–11, doi:10.1093/jas/skaa233), if it is possible, some of the data in J Anim Sci (i.e., antioxidant parameter) could be used to establish relationship with the microbial data in this study.”

Response to the reviewer: Thank you for this comment. We struggled with this exact issue, as it related to the markers of gastrointestinal function and integrity, and oxidative status. These data were already published in Petry et al., 2020 and therefore cannot be “re-published” herein. However, the authors have discussed a possible relationship of those data (and other literature) within the context of the data reported in this manuscript. Please refer to Lines 471-495.

Reviewer 2 Comments

General Comments. Revision: Line 110 

“ #1. Experimental design. A total of 60 growing gilts (progeny of Camborough sows × 337 sires; PIC Inc., Hendersonville, TN) were used in 3 replicates (20 gilts per replicate) of a 46-day trial. Gilts were individually housed in pens for 36 days, and subsequently moved to metabolism crates for the previously published 10-d metabolism study. So how many replicates/pigs are per treatment/replicate? It is not clear to readers.

Response to the reviewer: Thank you for pointing this out. There were 15 pigs per treatment, and 5 pigs per treatment within each of 3 replicates. This has been clarified in line 110 of the revised document. 

General Comments. Revision: 114-118

“ #2. Materials and methods: where are AX and xylanase from? Is AX extracted from corn bran? And also xylanase. Please note that different sources of AX exert different effects of microbial community composition.” 

Response to the reviewer: The AXOS used in this study are produced by hydrothermal treatment of corn cobs over an extended period; then they are treated with a xylanase to produce oligosaccharides which range from 3-7 degrees of polymerization and confirmed to have 95% purity. The xylanase supplemented in this study is a commercially available endo-beta 1,4-xylanase (Econase XT 25P; AB Vista, Marlborough, UK). It is produced using submerged fermentation by Trichoderma reesei. The xylanase used in this study has been previously shown to improve fiber digestibility of corn-based fiber (Petry et al., 2019; Weiland, 2017). This was a helpful comment, and this information has been added to Lines 114-118. 

General Comments. Revision: Line 397-398 

“ #3. Results: In the previous study by Zhao et al. (2019), Firmicutes and Proteobacteria were the two dominant phyla in the ileum digesta, which is in agreement with microbial community of the ileal mucosa in the current study. (Zhao et al. Fiber-rich foods affected gut bacterial community and short-chain fatty acids production in pig model.) However, how to explain differences in microbial composition between ileal digesta and ileal mucosa.

Response to the reviewer: Relative to the study conducted by Zhao et al., (2019), the diets used in this study are more complex and contain whole grains or co-product matrices. The corn bran diet in Zhao et al., (2019) is a semi-synthetic diet that contains protein isolates, sucrose, and isolated starch in addition to the fiber source, and is not directly comparable to the study herein. The more digestible diet in Zhao et al., (2019) would likely impact ileal microbiota differently as those ingredients are more soluble and readily fermentable, compared to complex grain matrices used herein. The differences between the microbial taxa observed in the digesta and mucosa found herein could be a result of differing oxygen levels, as the mucosa-associated microbiota is exposed to oxygen diffusion from the tissue. These mucosa-associated bacteria are often facultative anaerobes. This information has been added to the sentence in line 397-398.

General Comments. Revision: Line 88-89 

“ #4. Discussion: Xylanase plays positive actives in the foregut of pigs, such as decreasing viscosity of the intestinal digesta and increasing nutrient digestibility. However, AX, derived from corn and sorghum, is primarily fermented in the hindgut of pigs. Therefore, it is not reasonable to compare effects of xylanase and AX on microbial community in the ileum of the pig. Please discuss it.” 

Response to the reviewer: Thank you for bringing this point up. The authors agree that whole arabinoxylans associated with plant cell walls are predominately fermented in the hindgut where microbial density is greatest, and the xylanase exerts action in the small intestine. Interestingly, the soluble arabinoxylan oligosaccharides with short degrees of polymerization in other species appear to have a greater potential to be fermented by microbiota in the foregut (Tiwari et al., 2020). Part of the reason we focused on the ileum is because there is such a large paucity of research investigating AXOS in the small intestine of pigs but work in poultry has shown it can elicit a response early on in the gastrointestinal tract. We have made this clearer for the readers in L88-89 in the introduction. 

General Comments. Revision: Line 543-544

“ 5. Conclusion: The context in conclusion is redundant, and most of points are only conjectural. Such as Line 502-505, because no data for SCFA concentration was shown in the current study.

Response to the reviewer: The conclusions presented herein represent the key interpretations of the findings that are presented in the results and further considered within the discussion. For example, the conclusions stated in L533-509 refers to the discussion in L399-424. Likewise, the conclusions presented in L536-539 and L541-553 refer to the discussion in L510-531 and L471-497, respectively. Based on these examples of the foundation of our conclusions, we are confident that the conclusions presented in this manuscript are well supported by the data and the interpretation of those data. Moreover, the conclusions relate directly back to the objectives and hypothesis of the study presented in L90-97 of the introduction. However, we concur with the comment about SCFA and have altered the sentence to ‘favors microbial communities associated with the symbiotic cross-feeding of oligosaccharides derived from AXOS’ in L543-544. 

Specific Comments. Revision: Line 29 

“ #1 Line 29: 16S rRNA?”

Response to the reviewer: This has been corrected. 

Specific Comments. Revision: Line 51

“ #2. Line 49-53. Many recent study have reported prebiotic effects of dietary fiber on weanling pigs nutrition.”

Response to the reviewer: Thank you for this insight, and with regards to the context of this introductory paragraph, these statements are in reference to insoluble corn-based fiber that is poorly fermented and does not confer a prebiotic health benefit in the pig. This has been made clearer in Line 51. 

Specific Comments. No Revision 

“ #3. Line 353-360: This part described different responses of IDF and SDF on pig nutrition, but it is not relevance to objectives of the present study.”

Response to the reviewer: While this section of the discussion is surely more general in nature, it nevertheless provides the general background and conceptual framework behind our study. We would thus like to retain this section in the discussion, as DF and its effect on swine nutrition is the key aspect of this study. Moreover, these points underly all aspects of the gastrointestinal microbiota as diet is the key factor affecting microbial community composition.

Specific Comments. No Revision 

“ #4. Line 364-372: those contexts should be moved to the part of Introduction.

Response to the reviewer: These sentences provide context to the objective of the study and set the stage for subsequent discussion. This section of the discussion relates back to L73-75 of the introduction, and the authors believe moving these sentences to the introduction would be confusing to the reader.

---

## [Decision Letter · Decision Letter 1]

14 Jan 2021

Xylanase modulates the microbiota of ileal mucosa and digesta of pigs fed corn-based arabinoxylans likely through both a stimbiotic and prebiotic mechanism

PONE-D-20-30575R1

Dear Dr. Schmitz-Esser,

We’re pleased to inform you that your manuscript has been judged scientifically suitable for publication and will be formally accepted for publication once it meets all outstanding technical requirements.

Kind regards,

Francois Blachier, PhD

Academic Editor

PLOS ONE

Additional Editor Comments (optional):

Reviewers' comments:

Reviewer's Responses to Questions

**Comments to the Author**

1. If the authors have adequately addressed your comments raised in a previous round of review and you feel that this manuscript is now acceptable for publication, you may indicate that here to bypass the “Comments to the Author” section, enter your conflict of interest statement in the “Confidential to Editor” section, and submit your "Accept" recommendation.

Reviewer #1: All comments have been addressed

2. Is the manuscript technically sound, and do the data support the conclusions?

Reviewer #1: Yes

3. Has the statistical analysis been performed appropriately and rigorously? 

Reviewer #1: Yes

4. Have the authors made all data underlying the findings in their manuscript fully available?

Reviewer #1: Yes

5. Is the manuscript presented in an intelligible fashion and written in standard English?

Reviewer #1: Yes

6. Review Comments to the Author

Reviewer #1: This is a good piece of work. All the concerns have been addressed. I would like to recommend publication with issue highlihgts if possible.

7. PLOS authors have the option to publish the peer review history of their article (what does this mean?). If published, this will include your full peer review and any attached files.

Reviewer #1: **Yes: **De Wu, Sichuan Agricultural University

---

## [Editor Report · Acceptance letter]

18 Jan 2021

PONE-D-20-30575R1 

Xylanase modulates the microbiota of ileal mucosa and digesta of pigs fed corn-based arabinoxylans likely through both a stimbiotic and prebiotic mechanism 

Dear Dr. Schmitz-Esser:

I'm pleased to inform you that your manuscript has been deemed suitable for publication in PLOS ONE. Congratulations! Your manuscript is now with our production department. 

Kind regards, 

on behalf of

Dr. Francois Blachier 

Academic Editor

PLOS ONE